# Preliminary Virtual Constraint-Based Control Evaluation on a Pediatric Lower-Limb Exoskeleton

**DOI:** 10.3390/bioengineering11060590

**Published:** 2024-06-08

**Authors:** Anthony C. Goo, Curt A. Laubscher, Douglas A. Wajda, Jerzy T. Sawicki

**Affiliations:** 1Center for Rotating Machinery Dynamics and Control (RoMaDyC), Washkewicz College of Engineering, Cleveland State University, Cleveland, OH 44115, USA; a.goo@vikes.csuohio.edu; 2Department of Robotics, Michigan Engineering, University of Michigan Ann Arbor, Ann Arbor, MI 48109, USA; claub@umich.edu; 3Department of Health Sciences and Human Performance, College of Health, Cleveland State University, Cleveland, OH 44115, USA; d.a.wajda@csuohio.edu

**Keywords:** gait, exoskeletons, virtual constraint control, pediatric

## Abstract

Pediatric gait rehabilitation and guidance strategies using robotic exoskeletons require a controller that encourages user volitional control and participation while guiding the wearer towards a stable gait cycle. Virtual constraint-based controllers have created stable gait cycles in bipedal robotic systems and have seen recent use in assistive exoskeletons. This paper evaluates a virtual constraint-based controller for pediatric gait guidance through comparison with a traditional time-dependent position tracking controller on a newly developed exoskeleton system. Walking experiments were performed with a healthy child subject wearing the exoskeleton under proportional-derivative control, virtual constraint-based control, and while unpowered. The participant questionnaires assessed the perceived exertion and controller usability measures, while sensors provided kinematic, control torque, and muscle activation data. The virtual constraint-based controller resulted in a gait similar to the proportional-derivative controlled gait but reduced the variability in the gait kinematics by 36.72% and 16.28% relative to unassisted gait in the hips and knees, respectively. The virtual constraint-based controller also used 35.89% and 4.44% less rms torque per gait cycle in the hips and knees, respectively. The user feedback indicated that the virtual constraint-based controller was intuitive and easy to utilize relative to the proportional-derivative controller. These results indicate that virtual constraint-based control has favorable characteristics for robot-assisted gait guidance.

## 1. Introduction

A lower-limb exoskeleton is a wearable robotic device that provides assistive torque to the joints of the wearer’s legs. In medical contexts, exoskeletons can be used to assist or rehabilitate the motion of individuals dealing with gait impairment through robotic-assisted gait training (RAGT). RAGT has been suggested as an alternative or complementary solution to traditional physical therapy options and bodyweight-supported treadmill training. The introduction of a robotic device to guide the gait pattern decreases the physical demands on the physical therapist and offers increased robotic accuracy and controllability to the walking task [1,2]. Previous studies have shown that RAGT can increase the wearer’s average walking speed, distance, balance, and other mobility measures [3,4]. Studies have also demonstrated that RAGT can improve the range of motion, increase muscle strength, and decrease spasticity for pediatric subjects with cerebral palsy [5,6,7].

While children with gait impairments stand to benefit from RAGT, most commercially available exoskeletons are adult-oriented [8,9] and are not designed to serve the pediatric population [10]. Representative pediatric devices currently include the pediatric Lokomat [11], the Trexo robotic walker [12], the very small-sized Hybrid Assistive Limb (2S-HAL) [13], the ATLAS 2020 and 2030 [7,14], the MOTION exoskeleton by Zhang et al. [15], and the exoskeletons developed by Lerner et al. at the NIH [5]. Of the pediatric devices that do exist, few devices combine the characteristics of a lightweight form factor, community setting mobility, adjustability, and ease of use. Previously, the authors created an anthropometrically parametrized exoskeleton [16], and recently introduced the Cleveland State University (CSU) adjustable pediatric exoskeleton [17,18]. Preliminary human factor testing with the new device demonstrated that the exoskeleton was suitable for preliminary control testing with pediatric subjects [19].

Identifying an appropriate controller for medical exoskeletons remains a challenge, in large part due to the diversity of gait impairment pathologies. The therapeutic objective for those who need walking assistance due to severe neurological injury differs greatly from those seeking gait rehabilitation and guidance, such as individuals recovering from stroke [20]. In this manuscript, the authors wish to investigate controllers suitable for gait guidance and rehabilitation. A common strategy for exoskeleton control includes time-dependent, position tracking controllers such as proportional-derivative (PD) and proportional-integral-derivative (PID) controllers [21,22,23]. Closely related time-dependent controllers include impedance controllers, which improve human–robot interaction safety by introducing compliant behavior between the wearer and the exoskeleton through model-based control [24,25]. Relevant examples include the LOPES robot by van der Kooij et al. [26], the knee device by Aguirre-Ollinger et al. [27], and the impedance control law used by Tran et al. on the HUALEX [28]. These controllers oftentimes utilize nominal human walking patterns from sources like Winter et al. [29] or Schwartz [30], to define the desired joint motion reference and spatiotemporal gait parameters. However, while time-dependent trajectory tracking controllers are effective at matching a gait pattern and are easy to implement, the strict timing nature can disincentivize user participation in the walking cycle, leading to patient passivity [31,32]. This in turn can lead to less efficient therapy sessions and inconclusive rehabilitation results. Other manuscripts have noted that the strict regulation of gait, especially timing, can lead to gait destabilization [33]. This aligns with the “guidance hypothesis”, which predicts that feedback can negatively impact motor learning and rehabilitation when heavily relied upon to complete that learned action [34]. Additionally, these time-dependent trajectory tracking controllers also risk gait desynchronization between the walking cycle of the controller and the intended gait of the wearer. This often results in the user fighting the exoskeleton controller and can lead to gait instability and potential falls. Thus, while these controllers are useful for walking assistance purposes, they are oftentimes not suitable for gait rehabilitation.

The shortcomings of strict time-dependent position controllers for rehabilitation purposes have encouraged the exploration of patient-cooperative and time-independent controllers. A prominent example of these are the “virtual tunnel” controllers utilized by the ALEX [3] and Lokomat [35] exoskeletons. These controllers are designed to provide restorative inputs when a patient deviates from the desired gait pattern by a certain threshold. The ALEX’s force-field controller aims to guide the motion of a user’s ankle [3], while the Lokomat’s path control mode focuses on the overall leg posture through the gait cycle [35]. A continuation of this control methodology can be found in the paper by Martínez et al. [36], which utilizes force-field controllers to guide a lower leg exoskeleton during the swing phase. These controllers enable the wearer’s volitional control over the gait cycle and encourage their active participation in the walking activity. They have also been implemented in both time-dependent and -independent formulations. However, while the increased level of volitional control over the gait cycle encourages rehabilitation and user participation in the exercise, it only indirectly encourages a user’s dynamic stability during gait.

Recent advances in the control of bipedal robotic systems have yielded a new control methodology in virtual constraint-based controllers, also commonly referred to as hybrid zero dynamic controllers. These controllers enforce relationships between the system’s joints such that the biped walker becomes virtually constrained to walk in a certain pattern [37]. The strategic definition and optimization of these virtual constraints, which evolve with respect to the gait phase, can promote a dynamically stable gait for biped systems within their zero dynamics. These controllers can also be implemented in time-invariant formulations by representing the gait phase as a configuration-dependent variable. Extended to exoskeletal systems, these controllers drive the wearer towards the stable gait cycle defined by the virtual constraints, while leaving progression through the gait cycle dependent on the volitional control and effort exerted by the user. While originally implemented in fully robotic bipedal systems in [38,39,40,41], virtual constraint-based controllers have begun to see use in both prosthetic and exoskeletal devices [42,43,44]. Most of these applications, however, are focused on walking assistance for paraplegic patients instead of gait rehabilitation objectives [42,43]. To the author’s knowledge, there have been few studies looking to evaluate virtual constraint-based controllers for gait guidance and rehabilitation.

In this manuscript, the authors aim to preliminarily evaluate a virtual constraint-based controller for gait guidance by performing a comparison to time-dependent proportional-derivative control in treadmill walking experiments. Both the virtual constraint and proportional-derivative controllers utilize identical control gains and gait references to increase comparability. The two controllers are evaluated with respect to the kinetic and kinematic effects of the controllers on the subject’s gait, the subject’s muscle effort quantified through electromyography (EMG), and the subject’s perceived effort and controller preferences as indicated through questionnaires. The authors hypothesize the following:The two controllers will have comparable kinematics due to their similar error-based architecture and comparable control gains;The virtual constraint-based controller will demonstrate less gait pattern variability due to the lower risk of gait desynchronization;The virtual constraint-based controller will be preferred over the proportional-derivative controller due to its time-independent nature and lack of step timing restrictions.

This work builds upon the authors’ previous work on virtual constraint-based controllers for gait guidance [45,46] by applying a virtual constraint-based controller on a newly developed pediatric exoskeleton system with an able-bodied child subject. This manuscript demonstrates the adjustable pediatric lower-limb exoskeleton’s ability to serve as an investigative platform for future gait assistance and rehabilitation control experiments. Thus, the contributions of this manuscript are as follows:A preliminary evaluation of virtual constraint-based control for gait guidance by performing a comparison to a more commonly applied time-dependent proportional-derivative controller;The demonstration and first application of control on the CSU adjustable pediatric exoskeleton in gait experiments.

The successful implementation of virtual constraint-based control on the exoskeletal system for gait guidance purposes represents an initial motivating step towards larger-scale rehabilitative control studies involving children with gait disabilities. The remainder of the manuscript is split into the following sections. Section 2 details the materials and test facility used in the gait experiments performed in this work. Section 3 details the controller implementations used in this control comparison. Section 4 discusses the experimental procedure. Section 5 presents and discusses the experimental results. Finally, Section 6 consists of the conclusion and points out avenues for future work.

## 2. Hardware and Facilities

### 2.1. Adjustable Pediatric Lower-Limb Exoskeleton

The CSU adjustable pediatric exoskeleton provides supplementary torques at the hip and knee joints of the wearer through 144 W brushless DC motors, scaled through a 20.4:1 two-stage belt and chain transmission. The modular actuators can apply up to 5.9 Nm of continuous torque, have been tested to up to 21.1 Nm peak torque, and have a theoretical peak torque of 46.9 Nm. Previous evaluations indicated that the actuators were lightweight, low-friction, and easily backdrivable at the output, making them appropriate for use in a pediatric lower-limb exoskeleton [17]. These actuators were placed into an adjustable pediatric exoskeleton frame designed for children between 6 and 11 years old [18], resulting in the 4.72 kg exoskeleton shown in Figure 1.

The ranges of adjustability were determined from estimated limb lengths and widths of children within the target age group, derived from anthropometric averages [47] and census data [48]. For a more detailed discussion of the exoskeleton device and joint actuators, see [17,18]. A preliminary human factor assessment with the unpowered adjustable pediatric exoskeleton and a healthy, 30.8 kg, and 149 cm tall child volunteer subject demonstrated that the hardware was comfortable, easily adjustable, and simple to don and doff [19]. The exoskeleton can provide a measurement of the relative joint angles and velocities for the hips with respect to the torso and the knees with respect to the thigh for both legs through the Hall effect and magnetic angle sensors. A SEN-10736 (Sparkfun Electronics, Boulder, CO, USA) nine-degree-of-freedom inertial measurement unit (IMU) is affixed to the hip cradle to provide angular position and velocity measurements of the torso relative to the gravity vector. The measurement convention for the human–exoskeleton system is shown in Figure 2, with the clockwise rotations in the figure representing positive rotations.

### 2.2. Treadmill, Sensors, and Data Collection

The gait experiments in this manuscript were performed on an R-Mill instrumented split belt treadmill (Motekforce Link, Amsterdam, The Netherlands), shown in Figure 3.

The system includes parallel bar structures and an overhead harness suspension system to assist with the subject’s lateral balance and provide a safety precaution in case of a fall. The instrumented treadmill provides ground reaction force (GRF) signals as analog outputs from the force plates for both the left and right sides.

The subject’s muscle activations were measured through a Trigno wireless EMG system (Delsys Incorporated, Natirck, MA, USA). The authors measured activations in the Vastus Medialis (VM), the Rectus Femoris (RF), the Biceps Femoris (BF), the Tibialis Anterior (TA), the Gastrocnemius Medialis (GM), and the Gastrocnemius Lateralis (GL). The Vastus Lateralis was originally measured but the associated EMG sensor fell off mid-experiment, so the analysis on this muscle was excluded. Only the dominant leg of the subject was equipped with EMG sensors. The outputs of the EMG sensors were filtered through a second-order Butterworth bandpass filter between 30 and 300 Hz, full-wave rectified, and then low-pass filtered with a second-order Butterworth filter with a cutoff frequency of 20 Hz to yield the linear envelope of the signal.

The control and partial data acquisition for this experiment were facilitated through a dSPACE MicroLabBox DS1202 (dSPACE, Wixom, MI, USA). The dSPACE system collected the joint angle and velocity measurements from the exoskeleton, and GRF data from the instrumented treadmill. Separately, the wireless EMG sensors and GRF measurements were also recorded through D-Flow (version 3.34.3) and CORTEX (64-bit, version 8.1.0.2017) software to yield data files sampled at 10 kHz. These data sets were then unified in time by aligning both sets’ measurements of the GRF data.

## 3. Control Overview

This manuscript compares the performance of a time-dependent proportional-derivative (PD) trajectory tracking controller and a virtual constraint-based (VC) controller. An unassisted (UA) condition, with the subject walking in the unpowered exoskeleton, was also tested to serve as a baseline condition for comparison.

In the PD controller, the input torque is defined proportionally to the position and velocity error of the system relative to a reference gait pattern. The reference gait profiles for each joint were derived from the unassisted walking pattern of the wearer using the unpowered exoskeleton, taken on a previous testing day. This was chosen over nominal gait to represent the gait closest to the subject’s natural walking cycle while constrained by the movements allowed by the exoskeleton. This gait pattern served as the desired gait profile across all the controlled conditions. The proportional and derivative control gains applied to the system were chosen by the subject. For ease of implementation and for the sake of comparison, the same set of control gains were used across all the hip and knee joints and were used for both the PD and VC controllers. The time-dependent nature of the PD controller necessitated a method to align the controller step timing with the user. This was achieved by synchronizing a metronome to the gait period such that an audial cue was given to the subject on when to time their heel strike.

For a more in-depth review and stability analysis of VC-based controllers, the authors point the reader to the seminal works of Grizzle and Westervelt et al. [37,49]. In general, the virtual constraint-based control method generates a set of constraint functions $h(s\left(q\right))$ for the hip and knee joints that are dependent on a monotonically increasing phase variable s(q). This phase variable represents the progression of gait and is dependent on the configuration variable vector q. In prior experiments, the authors found that some phase variable definitions were sensitive to natural human gait variability, which led to unnatural human–exoskeleton behavior during control implementation [45]. Thus, the authors utilized a phase definition determined via optimization as performed in previous works, using the gait data from the UA condition [46]. This optimization identifies a phase variable definition of the form shown in Equation (1):(1)$$s\left(q\right)=cq+s_{0}$$

The row vector c and the constant $s_{0}$ reflect the set of constants identified through the optimization. The optimization of the phase definition was subject to the following constraints:(2)$$s\left(q^{-}\right)=0\;s\left(q^{+}\right)=1\;s^{\prime }\left(q_{i}\right)>0$$
and minimizes the cost function shown in Equation (3):(3)$$J_{s}=\sum_{i=1}^{N}{\left(s^{\prime }\left(q_{i}\right)-1\right)}^{2}$$

The expression of $q^{-}$ and $q^{+}$ represents the joint configuration vector of the human–exoskeleton model at the beginning and ending of a step, respectively, while $q_{i}$ represents the system configuration at a single datapoint 1≤i≤N. The expression for the phase rate with respect to normalized time $\hat{t}$ is denoted as $s^{\prime }=\Delta s/\Delta \hat{t}$. The result is an optimal phase definition that evenly distributes the phase’s sensitivity to natural human gait variability over the entire gait step and is roughly equivalent to normalized time.

This phase definition is then utilized in a second offline optimization that generates constraint functions $h(s\left(q\right))$. The optimization aligns the gait cycle described by the constraint functions with the gait cycle recorded from the UA baseline. The optimization is carried out via the Trajectory Optimization in CasADi (TROPIC 1.18.2021) toolbox [50], utilizing the cost function shown in Equation (4).
(4)$$J_{TROPIC}=\sum_{i=1}^{N}\left\|h\left(s\left(q\right)\right)-r_{i}\right\|$$

In the above equation, $r_{i}$ is the desired reference gait cycle recorded from the UA condition, where the subscripts denote the percent of step phase. Additional optimization constraints ensure that the walking speed of the optimized gait cycle matches the walking speed of the UA baseline, and that the step period was within two standard deviations of the baseline.

A time-invariant feedback controller then enforces the constraint functions. While theoretical implementations of virtual constraint-based controllers utilize feedback linearization controllers to demonstrate controller stability and convergence, practical implementations in simulation and hardware applications have demonstrated that phase-based PD works to drive the system towards the desired cyclical gait [38,49,51]. The control law used in this virtual constraint-based controller is shown below.
(5)$$Pe+D\dot{e}=u$$
(6)$$e=h\left(s\left(q\right)\right)-Hq$$
(7)$$\dot{e}=\frac{\partial h(s\left(q\right))}{\partial s}\dot{s}-H\dot{q}$$

The error vectors e and $\dot{e}$ represent the position and velocity errors of the system with respect to the virtual constraint functions. The matrix H consists of ones and zeros and maps the controlled joints of q and $\dot{q}$ to their appropriate constraint functions. Specifically, $H=\left[0_{4\times 1}\;I_{4\times 4}\right]$, where $I_{4\times 4}$ is a 4-by-4 identity matrix. P and D are positive diagonal gain matrices.

The VC controller relies on a pinned model of the human–exoskeleton system. However, human gait exhibits periods of double support and the swapping of stance and swing legs. In traditional virtual constraint-based controllers, the double support phase is modeled as an instantaneous impact event with a transformation of the system states and the swapping of the swing and stance leg definitions. In this paper, the bilateral mixing strategy from [45,46] is used. It defines two symmetric full-body controllers, $u_{l}$ and $u_{r}$, which assume that either the left or right leg are in the stance phase, respectively. The total control inputs to the system $u_{tot}$ are then defined as a convex combination of the two controllers, where the weights of the two controllers, $w_{l}$ and $w_{r}$, are based on GRF measurements. The bilateral mixing strategy is shown in Equation (8), and the definition of the weighting coefficients are defined in Equation (9), with respect to the left and right vertical GRF $f_{r}$ and $f_{l}$.
(8)$$u_{tot}=w_{l}u_{l}+w_{r}u_{r}$$
(9)$$w_{l/r}=\left\{\begin{array}{cc} \frac{f_{l/r}}{f_{l}+f_{r}}, & f_{l}+f_{r}\;≉0\\ 0, & f_{l}+f_{r}\;\approx 0 \end{array}\right.$$

This bilateral mixing strategy enables transitions across double stance phases without discontinuous control inputs. Additionally, it allows for automatic control switching between the two controllers when either the left or right leg is serving as the stance leg.

## 4. Experimental Procedure

An 11-year-old female volunteer subject participated in this study along with their adult caretaker. The subject weighed 30.8 kg and measured 149 cm in height. They had been exposed to the exoskeletal device through the previous human factor assessment [19]. The exoskeleton was comfortably compatible with the subject and was adjusted to their anthropometrics at the start of the experiment. The anthropometric parameters of the subject’s limbs were estimated from census data using their overall height and weight. The parameters of the exoskeleton system were manually measured or derived from CAD models of the system. The two models were combined to generate an approximate human–exoskeleton rigid body model, with parameters listed in Table 1 for the torso, thigh, and shank.

The volunteer participant was informed of the experiment’s motivations and purpose, and written assent and informed consent was given by both the subject and their parent/guardian prior to the start of the study in accordance with the Institutional Review Board at Cleveland State University. The procedure consisted of three sessions. The first session was to familiarize the subject with the placement of the EMG sensors, and to perform preliminary sensor and control calibrations. A research assistant modeled the placement of the EMG sensors on their own leg so that the parent/guardian could accurately place the sensors on the child’s limbs. Torque saturation limits were identified for each joint of the exoskeleton by having the subject maintain a neutral single stance standing position while a slowly ramping torque was applied to the joints of the non-stance leg. The subject indicated the upper limit of torque that they were able to overpower or resist from the exoskeleton. These torque limits were implemented as a safety precaution so that the wearer could forcibly exercise control over the gait cycle in case of controller desynchronization. Torque ranges from −5 to 8 Nm and from −4 to 4 Nm were identified for the subject’s hips and knees, respectively. Next, the subject walked on the treadmill while wearing the exoskeleton in an unpowered condition. After the subject became accustomed to walking with the exoskeleton, a set of gait data for unassisted walking was taken to serve as the baseline reference for the PD and VC control conditions.

The next session served as a practice day. The subject practiced walking with the exoskeleton in the unassisted, PD-, and VC-controlled conditions for 6 min each. This training day allowed the subject to learn how to walk with the exoskeleton under each control condition before data were recorded. This practice day was conducted to mitigate the temporary effects of the patient’s learning period during final data acquisition. During these early gait sessions, preliminary subject-selected control gains were identified as a starting point for the later gait experiments.

The third experimental session consisted of the final set of gait experiments and the collection of data and subject questionnaires. Each of the tested walking conditions started with a gait synchronization event. This allowed the researchers to synchronize the D-Flow and dSPACE data sets in time by aligning the GRF measurements during data processing. The treadmill system was sped up to a user-selected walking speed of 0.8 m/s. The controller inputs were then incrementally increased until the subject-selected gains for the control condition were reached. The subject walked for 3 min under the controlled condition. Afterwards, the control inputs and treadmill speed were ramped back down. Following each test condition, the subject was given a 3-min rest period, during which a questionnaire was completed to allow the subject to give feedback and rate their perceived physical effort using the Borg Rating of Perceived Effort scale [52]. On the day of the experiment, the control conditions were applied in the following order: PD, VC, and finally, UA. After all the tested conditions were completed, the subject was asked to rank the applied controllers based on their exertion and subjective personal preference from least to greatest.

## 5. Results and Discussion

The subject made no notes regarding discomfort while wearing the exoskeleton and did not indicate excessive levels of fatigue. There were no recorded trips or falls during testing. On the day of the experiments, the subject chose control gains that produced conservative control inputs. The proportional and derivative gains for both the PD and VC conditions were left at 7.8 Nm·rad^−1^ and 0.12 Nm·s·rad^−1^ across both the hip and knee joints.

The gait information was partitioned into step cycles based on the GRF information such that the beginning and end of each gait cycle corresponded with the heel strike event. The left and right leg gait cycles were combined for the kinematic analysis. Only the right leg information was used for the EMG sensors analysis, as only the patient’s dominant leg was equipped with sensors.

### 5.1. Kinematics and Kinetics

To compare hip and knee angles across the conditions, a one-way analysis of variance (ANOVA) statistical test was performed. This was performed by comparing the hip and knee joint angles in each tested condition (PD, VC) to that of the UA baseline and taking the average root mean square (rms) difference. Each comparison was performed using a *t*-test (significance level of 0.050) with a Tukey–Kramer multiple-comparisons correction. Figure 4 illustrates the ensemble-averaged gait cycle accomplished under the different controlled conditions, plotted with respect to the UA condition performance.

Table 2 lists the quantified performance metrics averaged over the gait cycle such as the mean rms difference with respect to the UA baseline, their mean standard deviations, and the mean rms torque output.

Low angular differences were recorded across the controlled conditions for both the hips and knees. The PD-controlled condition reported an rms difference of 2.50 and 3.75 degrees in the hips and knees, respectively, while the VC-controlled condition reported slightly higher differences of 3.06 and 4.59 degrees for the hips and knees. The rms differences for both the hip and knee positions relative to the UA baseline were sufficiently similar such that statistically significant (*p* < 0.05) differences were not identified between the PD vs. VC conditions in the 275 gait cycles compared. This indicates that the gait cycles in the PD- and VC-controlled conditions were comparable despite the difference in the controller used. This is further corroborated when looking at the effect size between the conditions, which are listed in Table 3.

For all the comparisons (UA-PD, UA-VC, and PD-VC), the differences in the kinematics were moderate with an average absolute effect size within 0.48–0.67 for the hip and 0.47–0.58 for the knee. This suggests the gait patterns are largely similar, often within a standard deviation of one another across all the pairs of conditions. In the context of the experiment performed, this is unsurprising, as the subject was a healthy individual and the control gains were tuned such that the subject could manually exert control over the gait cycle. However, while the average gait profiles in each condition were similar, a point-wise calculation of the standard deviation was obtained and then averaged to quantify the gait variability. The mean standard deviation of the hip and knee angles decreased in the VC condition relative to the UA and PD conditions. The VC controller decreased the wearer’s gait variability from 4.88 to 3.04 degrees in the hip and 6.33 to 5.30 degrees in the knee between the UA and VC conditions. This represents a relative reduction in the mean standard deviation at the hip and knee joints of 36.72% and 16.28%, respectively. In the PD controller, the mean standard deviation of the hip joints decreased to 3.70, or only 27.03%, and for the knee, increased to 6.78 degrees, representing a 7.10% increase. These changes in the standard deviation indicate that the VC controller increased the wearer’s gait regularity and consistency more than the PD controller.

An additional ANOVA and multiple comparisons *t*-test was performed on the rms torque profiles of each controlled condition to quantify the changes in the amount of applied intervention. There was a statistically significant (*p* < 0.050) reduction in the rms torques applied by the VC controller relative to the PD controller in the hip and knee joints. With regard to the ensemble-averaged torque profiles applied, the VC controller reduced the rms torques applied from 0.39 to 0.25 Nm and 0.90 Nm to 0.86 Nm for the hip and knee joints relative to the PD controller, representing a 35.89% and a 4.44% reduction in the overall robotic intervention, respectively. The kinematic and kinetic data indicate that the mean gait cycles of the VC- and PD-controlled conditions were similar, but the VC controller demonstrated a greater degree of gait regularity in the subject’s walking pattern while using less robotic intervention.

### 5.2. EMG Sensors and Perceived Exertion

Only the right leg of the subject was equipped with EMG sensors, which means the total number of gait cycles available for analysis was around half of those used in the kinematic data analysis. A total of 136 gait cycles were compared between the UA, PD, and VC conditions for the EMG analysis. Before the analysis, the EMG signals of each muscle were normalized with respect to the mean output of the muscle during the UA condition. Figure 5 plots the normalized mean and standard deviation of the EMG readings for each muscle group measured across all the tested conditions, while Figure 6 plots their normalized value over the gait cycle. These values are also listed in Table 4, along with Borg scale ratings and the post-experiment exertion rankings provided by the child subject.

To quantify the differences in the muscle activation levels, a similar statistical analysis was performed on the normalized EMG outputs for each muscle. Statistically significant differences in the muscle activation levels were found in the VC vs. PD and the VC vs. UA comparison, but not in the UA vs. PD comparison. The general trends demonstrate that the UA condition required the least amount of physical exertion based on the EMG measurements. The second lowest average EMG outputs were measured for PD, at about 11.8% higher than those of UA, followed by VC, at 34.4%.

The analysis of the child subject’s ratings of perceived exertion show some inconsistencies, and a discrepancy between the muscle activation levels and controller preference. The Borg scale ratings given after each tested condition indicates that the extent of perceived exertion between the PD- and VC-controlled conditions were similar. The PD-controlled condition was initially listed as a slightly lower effort controller than the VC condition, though this perception may have been affected by the order in which the controlled conditions were applied. The PD controller was applied before any other control conditions, and thus, there is likely some recency bias associated with the Borg scale rankings. The post experiment user rankings given at the end of all the gait experiments indicate that the user found the VC controller preferable to PD. These results stand in opposition to the fact that the muscle activations in the VC controller are higher than those in the PD-controlled condition. While more effort was expended by the user to walk under the VC-controlled condition, the controller was still seen as preferable to the PD controller. A potential explanation for this discrepancy is that the final rankings of preference in these experiments might serve more as a measure of ease of use, or how intrusive the controller was with respect to the wearer’s gait. For instance, the time dependency of the PD controller dictates a certain rate of gait progression and step timing. If the subject’s intended gait is lagging or leading the PD controller’s reference, this could result in the controller pushing or resisting the motion of the user. Gait desynchronization could lead to the user fighting the controller at certain points in the gait profile even if the PD controller is working cooperatively with the wearer for most of the motion. There were a few instances of gait desynchronization between the user and PD controller during the experiment, including a few gait cycles where the user and controller were completely desynched. Additionally, the standard deviation of the torque curve in Figure 4 is noticeably larger than that of the VC-controlled condition, suggesting a greater degree of variance in the control inputs, which themselves stem from how well the user’s gait is synchronized to the controller. In contrast, the VC-controlled condition leaves the gait timing entirely up to the wearer’s volition. While these controllers may not push the user through the gait cycle at any point, they never actively resist the intention of the wearer.

Across all the controlled conditions, the EMG outputs for each muscle increased relative to the UA case. Similarly, the Borg scale ratings and the finalized rankings of the tested conditions indicated that the subject found it more difficult to walk with external controllers than without. This suggests that for able-bodied subjects, the introduction of control inputs in this experiment acted more as a system disturbance as opposed to a restorative or assistive force.

### 5.3. Study Limitations

There are a few limitations to the conducted study. This study utilized only a single volunteer pediatric subject, which limits the generalizability of these results to other individuals. Additionally, the control inputs applied by the exoskeleton remained low. The average peak torque input was in the VC condition at merely 2.91 ± 0.63 Nm at the knee. This represents around 15.36% of the 18.94 Nm peak knee torque expected based on Winter gait data for a 30.8 kg subject [29]. The low control input can be attributed to the fact that the subject did not have any form of gait impairment, and so the amount of control input exerted by the exoskeleton remained low throughout the walking cycle. Additionally, the control gains applied in the experiment were tuned based on user comfort, resulting in gains that minimally affected the already well-performing gait cycle. However, repeated exposure to the exoskeleton and controllers may encourage the user to adopt a more cooperative walking strategy, or increase the subject’s confidence in the device, leading to the use of higher control gains and inputs. Additionally, a user with gait impairment may be more amenable to increased robotic intervention.

For similar reasons, the interpretation of the EMG-measured outputs should be taken with caution. While statistically significant differences in EMG activations were found between the controlled conditions, it is unclear whether the comparisons and relationships discussed in this paper will hold true when greater control inputs are allowed by a healthy individual or a user with gait impairment.

## 6. Conclusions

This paper presented a comparison of a virtual constraint-based controller with a traditional proportional-derivative controller to evaluate their suitability for gait guidance. Control was applied on a newly developed adjustable pediatric exoskeleton, marking the device’s first use in a control study. During the experiment, the subject gave no indication of discomfort due to the applied controller or the physical hardware. The authors successfully conducted multiple gait experiments using the exoskeleton under different controllers. The successful control implementation demonstrates that the adjustable pediatric lower-limb exoskeleton may serve as a platform in future experiments on rehabilitative and assistive controllers for children.

The virtual constraint-based controller achieved similar levels of gait performance relative to the proportional-derivative controller, as evidenced by the moderate effect size values. However, the VC controller was able to decrease the level of gait variability by 36.72% and 16.28% for the hips and knees, respectively. Conversely, the PD controller decreased variability in the hip joint by only 27.03% and increased the gait variability in the knee joint by 7.10%. Additionally, the VC controller utilized 35.89% and 4.44% less torque in the hip and knee joints relative to the PD controller. A comparison of the EMG outputs between the two controllers indicated that the virtual constraint-based controller required more effort to utilize. However, the user’s post-experiment controller rankings indicated that the VC-based controller was easier to utilize. This could be attributed to the difference in time dependence between the PD and VC controllers, which is evidenced by both the large standard deviation in the control torque inputs and the observations of gait desynchronization made by both the wearer and authors during the PD-controlled experiment.

The results of this study’s comparison suggest that virtual constraint-based controllers have favorable characteristics relative to standard PD control due to their perceived ease of use, decreased gait variability, and ability to reduce the control torque required to achieve good performance all while maintaining a time-invariant control implementation. The VC controller also allows the user to retain volitional control over the step timing and removes the risk of gait desynchronization during walking. Thus, virtual constraint-based controllers merit further investigation in larger multi-subject rehabilitation-oriented studies. The efficacy of virtual constraint-based controllers for rehabilitation should also be evaluated through the application of control on a pediatric subject dealing with gait impairment. The authors also propose a multi-subject study utilizing the newly validated exoskeleton platform to better demonstrate the exoskeleton’s ability to adjust to several pediatric subjects.

## Figures and Tables

**Figure 1 bioengineering-11-00590-f001:**
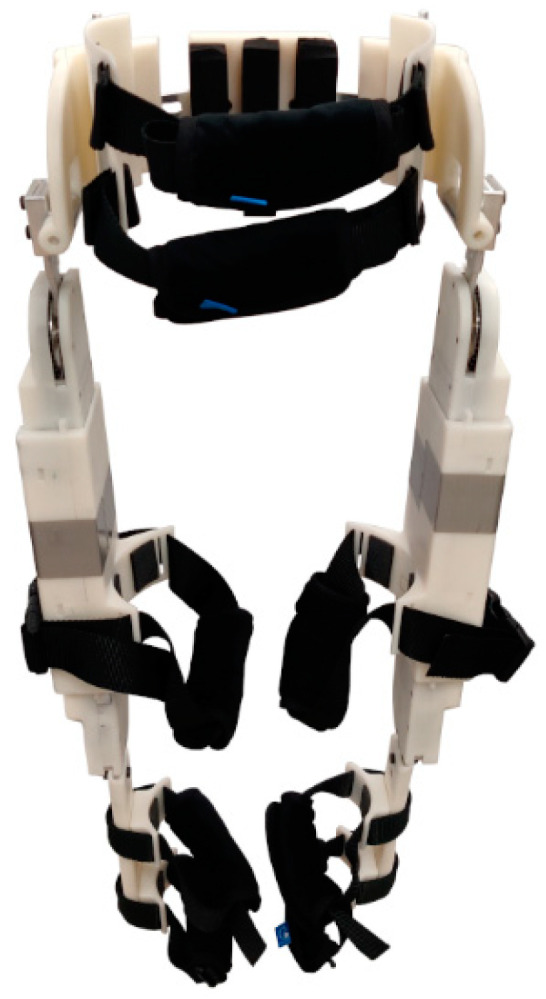
CSU adjustable pediatric exoskeleton.

**Figure 2 bioengineering-11-00590-f002:**
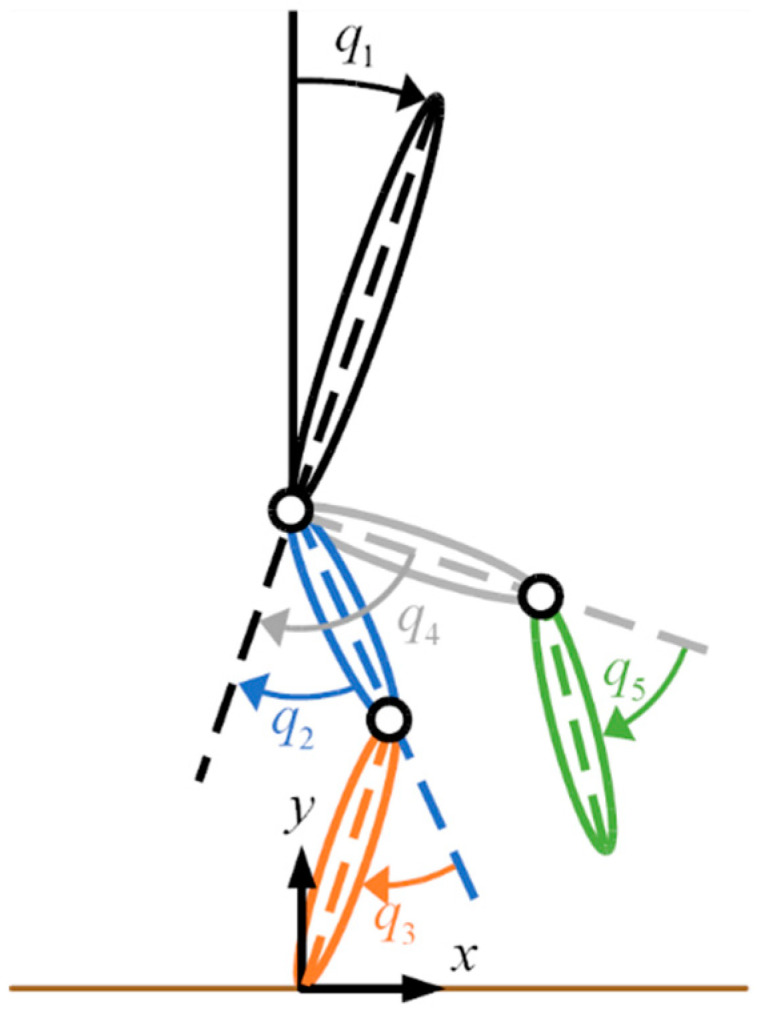
The measurement convention for the exoskeleton system and the controllers discussed. Hip extension and knee flexion correspond to positive values. The horizontal brown line denotes the location of the ground.

**Figure 3 bioengineering-11-00590-f003:**
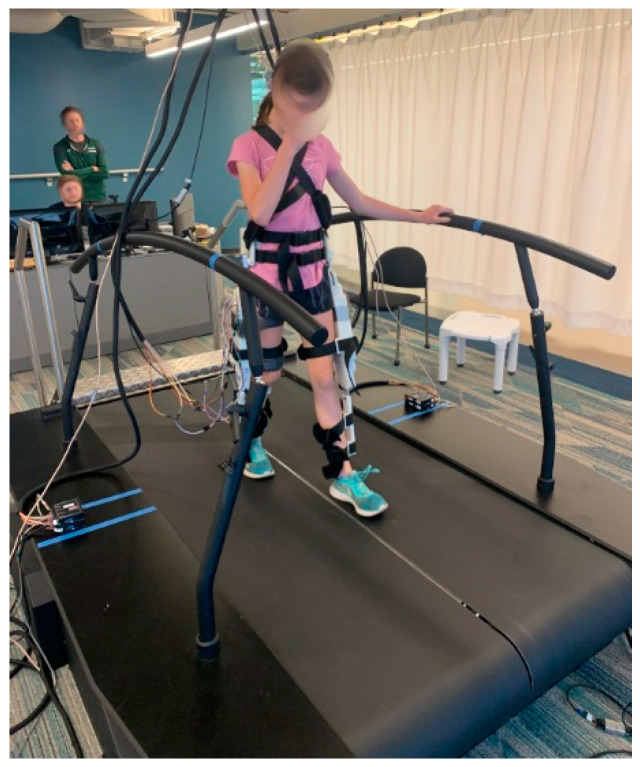
Experimental setup on the instrumented treadmill with the volunteer subject wearing the pediatric exoskeleton.

**Figure 4 bioengineering-11-00590-f004:**
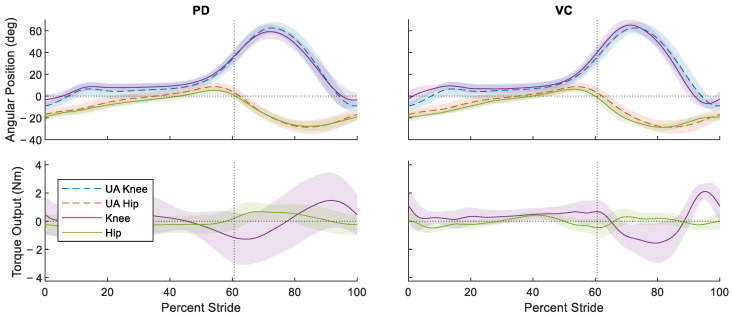
Ensemble-averaged position performance and torque input of the hip and knee joints for the proportional-derivative (PD) and virtual constraint-based (VC) control. They are plotted with respect to the joint performance of the unassisted (UA) baseline condition. The shaded regions show ±1 standard deviation for both position performance and control inputs. The vertical dotted line denotes the approximate location of toe-off.

**Figure 5 bioengineering-11-00590-f005:**
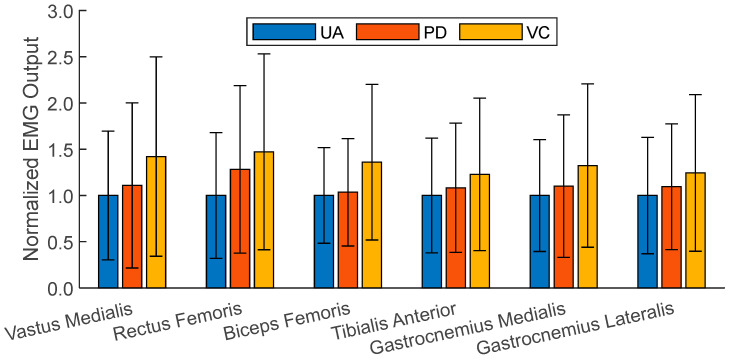
The normalized mean EMG outputs for each muscle and condition tested. The normalized standard deviations for the EMG outputs are represented as error bars.

**Figure 6 bioengineering-11-00590-f006:**
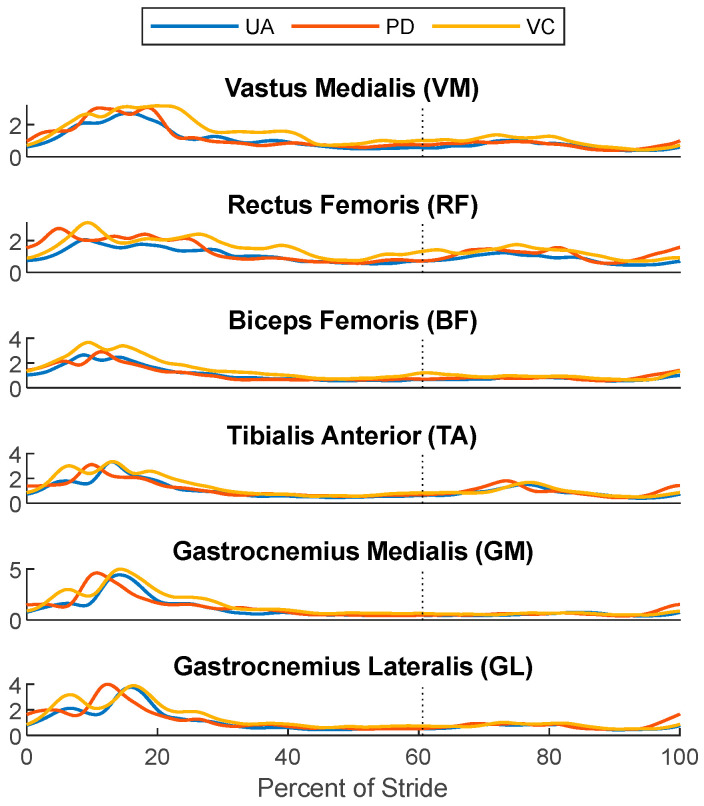
The normalized ensemble-averaged EMG outputs over the gait cycle, starting and ending with heel strike. The vertical dotted line denotes the approximate location of toe-off.

**Table 1 bioengineering-11-00590-t001:** Parameters of the human–exoskeleton system.

Link	Mass (kg)	Length (m)	CoM (m)	Inertia (kg·m^2^)
Torso	21.89	0.70	0.42	2.63
Thigh	4.51	0.36	0.16	0.06
Shank	2.12	0.42	0.25	0.06

Center of mass locations (CoM) are reported as distances along the body segments’ axial length with respect to the proximal joint. Inertia is reported with respect to the link center of mass.

**Table 2 bioengineering-11-00590-t002:** Performance and torque outputs of the control conditions.

Condition	Kinematic Difference *		Kinetics	
	Hip (deg)	Knee (deg)	Hip (Nm)	Knee (Nm)
UA	– ± 4.88	– ± 6.33	–	–
PD	2.50 ± 3.70	3.75 ± 6.78	0.39	0.90
VC	3.06 ± 3.04	4.59 ± 5.30	0.25	0.86

* Mean ± standard deviation. UA only reports standard deviation. – Represents a null entry.

**Table 3 bioengineering-11-00590-t003:** Effect size from control comparisons.

Comparison	Hip (Mean + Std)	Knee (Mean + Std)
UA-PD	0.4833 ± 0.3016	0.4766 ± 0.2302
UA-VC	0.6616 ± 0.2877	0.5769 ± 0.3535
PD-VC	0.5488 ± 0.2893	0.4765 ± 0.3070

UA, PD, and VC represent the unassisted, proportional-derivative, and virtual constraint-based controlled conditions respectively. Hyphens denotes the pairwise comparisons between conditions.

**Table 4 bioengineering-11-00590-t004:** Ratings of perceived exertion and normalized EMG outputs.

Condition	Borg	Rank	VM	RF	BF	TA	GM	GL
UA	7	1	1.00	1.00	1.00	1.00	1.00	1.00
PD	8	3	1.10	1.28	1.06	1.09	1.10	1.10
VC	9	2	1.43	1.46	1.36	1.24	1.32	1.25

Vastus Medialis (VM), Rectus Femoris (RF), Biceps Femoris (BF), Tibialis Anterior (TA), Gastrocnemius Medialis (GM), Gastrocnemius Lateralis (GL).

## Data Availability

The data presented in this study are available on request from the corresponding author.

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
