# Peer review of "Preliminary Virtual Constraint-Based Control Evaluation on a Pediatric Lower-Limb Exoskeleton"

_bioengineering, 2024, doi:10.3390/bioengineering11060590_

Round 1
Reviewer 1 Report
Comments and Suggestions for Authors
This paper evaluated the virtual constraint-based control for rehabilitation and gait guidance on a previously developed pediatric lower-limb exoskeleton with an able-bodied child subject. This topic is of great importance and has practical value for individuals with gait impairment. In general, the paper is clear and well organized.
Here, I have some concerns about this submission.
This work is about pediatric gait rehabilitation control strategies. However, the experiments were carried out with a healthy child subject instead of pediatric one. The authors themselves mentioned that the for able-bodied subjects the introduction of control inputs in this experiment acted more as a system disturbance as opposed to a restorative or assistive force. In fact, such an experiment design could be used in any kind of gait impairment (e.g., due to stroke, surgery, etc.)?
Besides, as the authors mentioned, one of the limitations in this work lies in the number of subjects (only one healthy individual was included), which limits the generalizability of the results.
These two problems make the effectiveness of the controller aimed at pediatric gait rehabilitation less convincing.
There are few minor points should be revised or described.
- In line 51, the abbreviation “CSU” is suggested to give the full name.
- In line 85, the word “ focusses” should be “focuses”.
- In line 113, “ with respect” should be “with respect to”.
- In table 4, row 2 column 4, “1.0” should be “1.00”.
- In line 253, the matrix H is suggested to be explained more clearly.
- Why for kinematics and kinetics comparison (in line 362) there are 275 gait cycles compared while for EMG sensors and perceived exertion (in line 394) there are 136 gait cycles?
Comments on the Quality of English LanguageThis paper evaluated the virtual constraint-based control for rehabilitation and gait guidance on a previously developed pediatric lower-limb exoskeleton with an able-bodied child subject. This topic is of great importance and has practical value for individuals with gait impairment. In general, the paper is clear and well organized.
Here, I have some concerns about this submission.
This work is about pediatric gait rehabilitation control strategies. However, the experiments were carried out with a healthy child subject instead of pediatric one. The authors themselves mentioned that the for able-bodied subjects the introduction of control inputs in this experiment acted more as a system disturbance as opposed to a restorative or assistive force. In fact, such an experiment design could be used in any kind of gait impairment (e.g., due to stroke, surgery, etc.)?
Besides, as the authors mentioned, one of the limitations in this work lies in the number of subjects (only one healthy individual was included), which limits the generalizability of the results.
These two problems make the effectiveness of the controller aimed at pediatric gait rehabilitation less convincing.
There are few minor points should be revised or described.
- In line 51, the abbreviation “CSU” is suggested to give the full name.
- In line 85, the word “ focusses” should be “focuses”.
- In line 113, “ with respect” should be “with respect to”.
- In table 4, row 2 column 4, “1.0” should be “1.00”.
- In line 253, the matrix H is suggested to be explained more clearly.
- Why for kinematics and kinetics comparison (in line 362) there are 275 gait cycles compared while for EMG sensors and perceived exertion (in line 394) there are 136 gait cycles?
Reviewer 2 Report
Comments and Suggestions for Authors
The paper evaluates a virtual constraint based controller for pediatric gait guidance through comparison with a traditional time-dependent position sensing controller on a newly developed exoskeleton system. The trial showed measures of perceived effort and controller usability, while sensors provided data on kinematics, control torque and muscle activation. The virtual constraint-based controller produced a similar gait to the proportionally-derivative controlled gait, but improved kinematics by both reducing variability in gait kinematics and used 35.89% and 4.44% less torque in the hip and knee respectively.
Finally, the authors argue that virtual constraint-based control has favorable characteristics for robot-assisted gait guidance.
The work is well organised, from the long and exhaustive introduction, through which the authors demonstrate an excellent knowledge of the state of the art, to the description of the results and conclusion.
The control was applied on a newly developed adjustable pediatric exoskeleton. The authors demonstrate that the VC controller was able to reduce the level of gait variability of 36.72% and 16.28% for the hips and knees respectively. This result is not very clear; this should be clarified better. The reduction of torque in the hip and knee also needs to be further clarified. Implementation of time-invariant control offers many advantages, as suggested by the authors, which need to be verified better (I find I have some doubts).
I suggest to include this paper about the lower variability in kinematics.
Huang, Z., Xi, F., Huang, T., Dai, J.S., Sinatra, R. Lower-mobility parallel robots: Theory and applications (2010) Advances in Mechanical Engineering, 2010, art. no. 927930. DOI: 10.1155/2010/927930
Round 2
Reviewer 1 Report
Comments and Suggestions for Authors
The authors explained some of my concern. Thanks.